

# TLR7 promotes skin inflammation via activating NFκB-mTORC1 axis in rosacea

Yaqun Huang[1,2,3,*], Da Liu[1,2,3,*], Mengting Chen[1,2,3], San Xu[1,2,3], Qinqin Peng[1,2,3], Yan Zhu[1,2,3], Juan Long[4], Tangxiele Liu[1,2,3], Zhili Deng[1,2,3], Hongfu Xie[1,2,3], Ji Li[1,2,3], Fangfen Liu[1,2,3] and Wenqin Xiao[1,2,3]

[1] Department of Dermatology, Xiangya Hospital, Central South University, Changsha, Hunan, China
[2] Hunan Key Laboratory of Aging Biology, Xiangya Hospital, Central South University, Changsha, China
[3] National Clinical Research Center for Geriatric Disorders, Xiangya Hospital, Central South University, Changsha, China
[4] Department of Dermatology, Hunan Children's Hospital, Changsha, Hunan, China
[*] These authors contributed equally to this work.

Corresponding authors
Fangfen Liu, 776407936@qq.com
Wenqin Xiao,
xiaowenqin0121@126.com

## ABSTRACT

Rosacea is a chronic inflammatory skin disease originated from damaged skin barrier and innate/adaptive immune dysregulation. Toll-like receptors (TLRs) sense injured skin and initiate downstream inflammatory and immune responses, whose role in rosacea is not fully understood. Here, via RNA-sequencing analysis, we found that the TLR signaling pathway is the top-ranked signaling pathway enriched in rosacea skin lesions, in which TLR7 is highlighted and positively correlated with the inflammation severity of disease. In LL37-induced rosacea-like mouse models, silencing TLR7 prevented the development of rosacea-like skin inflammation. Specifically, we demonstrated that overexpressing TLR7 in keratinocytes stimulates rapamycin-sensitive mTOR complex 1 (mTORC1) pathway *via* NFκB signaling. Ultimately, TLR7/NFκB/mTORC1 axis promotes the production of cytokines and chemokines, leading to the migration of CD4+T cells, which are infiltrated in the lesional skin of rosacea. Our report reveals the crucial role of TLR7 in rosacea pathogenesis and indicatesa promising candidate for rosacea treatments.

## INTRODUCTION

Rosacea, as a chronic inflammatory skin disease, is characterized by single or combined symptoms including erythema, telangiectasia, papules, pustules, edema and sensitive symptoms (such as stinging and burning), usually presenting in central face (*Van Zuuren & Rosacea, 2017*; *Xie et al., 2017*). Epidemiological survey revealed an approximately 5.5% prevalence of rosacea worldwide, which was higher among women between 45–60 years (*Gether et al., 2018*; *Li et al., 2020*). Apart from the physical and mental burden, rosacea also imposes an extra economic load on patients due to its high relapse rate and difficulty in etiological treatment resulted from undefined molecular mechanisms (*Van Zuuren & Rosacea, 2017*; *Thiboutot et al., 2020*; *Deng et al., 2018*).

The knowledge of rosacea so far indicates that genetic and environmental factors converge to induce and accelerate the process of disease through innate and adaptive immune dysregulation. As the body's first major line of defense, skin cells like keratinocytes respond to external dangers and interact with other cells to initiate defensive mechanism (*Nestle et al., 2009*). After the damage of skin barrier, trigger factors (including ultraviolet light, heat, physical or chemical stimuli, demodex and microbes) not only activate innate immune system, such as keratinocytes releasing epidermal antimicrobial peptide, mast cells releasing matrix metalloproteinase, macrophages and neutrophils releasing cytokines and chemokines, but also activate adaptive immune response, such as T helper cells (Th1, Th17), which eventually leads to the occurrence of rosacea (*Steinhoff et al., 2011*; *Buddenkotte & Steinhoff, 2018*; *Steinhoff, Schauber & Leyden, 2013*). However, the precise mechanism how impaired physical barrier initiates excessive immune regulation in rosacea calls for profound investigation.

As one of the best-studied and largest families of pattern recognition receptors, toll-like receptors (TLRs) identify specific microbial products or host damage products to initiate downstream inflammatory and immune responses (*Barton & Kagan, 2009*). Previous studies have demonstrated that excessively activated TLRs participate in numerous inflammatory diseases such as psoriasis, atopic dermatitis and so on (*Lai & Gallo, 2008*). For example, activated TLR2 was reported to elevate rosacea-related kallikrein 5 (KLK5), subsequently producing the pro-inflammatory forms of antimicrobial peptide LL37, and LL37 interacts with TLR2 to activate mTOR signaling, eventually inducing the development of rosacea (*Deng et al., 2021a*; *Yamasaki et al., 2007*). However, the understanding of TLR2 in rosacea is limited apart from these previous studies, so as other TLRs. TLR7 and TLR8, located in endosomes of human plasmacytoid dendritic cells (pDCs), could detect single-stranded RNA (ssRNA) from viruses or mammals (*Krieg, 2007*). They are able to sense injured skin by recognizing the complex of LL37 and self-derived nucleic acids released from damaged tissue and thereby stimulate Th1/Th17 T cell response in psoriasis or promote wound healing (*Ross et al., 2021*; *Hänsel et al., 2011*; *Gregorio et al., 2010*; *Ganguly et al., 2009*). Therefore, we wonder if TLR7 or TLR8 expressed by specific skin cells are involved in the pathogenesis of rosacea.

In this study, we reported that the TLR signaling pathway is significantly enriched in rosacea skin lesions. Among these TLRs, TLR7 is overexpressed in both rosacea patients and mouse models. Silencing TLR7 prevented the development of rosacea in LL37-induced rosacea-like mouse models. Mechanistically, we demonstrated that hyperactivated TLR7 in keratinocytes activates NFκB signaling, then stimulates rapamycin-sensitive mTOR complex 1 (mTORC1) pathway. Eventually, the TLR7/NFκB/mTORC1 axis promoted the production of rosacea-correlative cytokines and chemokines, leading to the migration of CD4+ T cells, which is involved in the process of rosacea development. Collectively, our observations revealed that TLR7 is essential in rosacea development and has the proficiency to be a therapeutic target of rosacea.

## MATERIALS AND METHODS

### Human samples

All the skin biopsies were obtained from the central face of 24 female patients (20–50 years old) diagnosed with rosacea by clinical and pathologic examination and 17 age-matched female healthy volunteers from the Department of Dermatology in Xiangya Hospital, Central South University. The patients details were shown in the supplementary materials of our previous study (*Deng et al., 2021a*). The severity of rosacea was evaluated by Investigator Global Assessment (IGA) scores and Clinician's Erythema Assessment (CEA) scores according to previous study (*Wang et al., 2021*). The utilization of all human samples was approved by the ethical committee of the Xiangya Hospital of Central South University (IRB number 201404361), and all the participants signed the written informed consent. This program was conducted in accordance with the WMA Declaration of Helsinki and the Department of Health and Human Services Belmont Report.

### Mice and treatments

Six weeks of BALB/c mice were purchased from Slack Company and bred under SPF conditions. The LL37-induced rosacea-like mouse model was generated as previously described (*Yamasaki et al., 2007*). Simply put, mice, anaesthetized by intraperitoneal injection of avertin, were shaved a day before injection and then were injected with LL37 peptide (40 μl, 320 μM) or control vehicle (PBS) on back skin twice daily for 2 days. The skin inflammation of mouse models was evaluated by the severity of erythema and edema according to previous methods (*Chen et al., 2019*). The redness was scored ranging from 1–5, and 5 being the reddest. The area of redness was measured by stereomicroscope measurements (Leica S8AP0; Leica, Wetzlar, Germany). Mice were anaesthetized and decapitated to obtain the back skin for histological Analysis, immunohistochemistry, immunofluorescence and RT-qPCR. For TLR7 knockdown treatment, 6 weeks of BALB/c mice were intradermally injected with 1.5 nmol/L scrambled/*Tlr7* siRNA (obtained from Shanghai GenePharma Co., Ltd.) three days and one day before LL37 injection. Procedures performed were approved by the ethical committee of the Xiangya Hospital, Central South University (IRB number 201611610).

### RNA-sequencing

Total RNA from human skin and HaCaT keratinocytes were extracted using TRIzol reagent (Thermo Fisher Scientific, Waltham, MA, USA). Each group of samples were analyzed as three biological replicants. Library preparation and transcriptome sequencing were prepared using Illumina HiSeq X Ten (Novogene, Beijing, China). The mapping of 100-bp paired-end reads to genes was conducted with HTSeq v0.6.0. The fragments of every kilobase of transcript per million fragments mapped (FPKM) were analyzed, and differential expression analysis was produced by the DESeq R package (1.10.1). The hierarchical clustering heat map was generated with the ggplot library. Differential expression data were ranked through $log_2$ fold change and performed Preranked GSEA to figure out enrichment for KEGG pathways (*Subramanian et al., 2005*). The statistical power of this experimental design, calculated in RNASeqPower is 1.0. The sequencing

depth is 37.00 (https://doi.org/doi:10.18129/B9.bioc.RNASeqPower). The nominal *P* value was adjusted with the Benjamini and Hochberg methods to correct the output results for multiple comparisons. The raw sequence data of HaCaT keratinocytes have been deposited in Figshare that are publicly accessible at https://doi.org/10.6084/m9.figshare.22725614.v2. Sequencing data from rosacea patients have been deposited in the genome sequence archive under accession number HRA000378.

## Histological analysis

Histological analysis was performed as previously described (*Wu et al., 2018*; *Zhao et al., 2018*). Skin samples of mice were fixed in formalin and then embedded in paraffin. Skins were cut into 5 μm sections and then were stained with hematoxylin and eosin (H&E). The number of infiltrating cells in dermis was identified as histological feature. The average number of infiltrating cells in six randomly selected microscopic areas (original magnification, 200×) in each sample was determined as the amount of infiltrating cells in dermis.

## Immunohistochemistry

Skin samples of human were fixed in formalin and then embedded in paraffin. Skins were cut into 5 μm sections. Immunohistochemistry was carried out according to previous methods (*Deng et al., 2019*). Skin sections were incubated with primary TLR7 antibody (1:100, Abcam). Pictures were captured from six typical areas in each sample.

## Immunofluorescence

Immunofluorescence of mice skin sections was carried out as previously described (*Tang et al., 2016*). In short, mouse skin samples were fixed in 4% paraformaldehyde (PFA) and frozen in OCT. After being cut into 8 μm, skin sections were washed with phosphate-buffered saline (PBS) three times and then blocked by blocking buffer (5% NDS, 1% BSA, 0.3% Triton X-100) for 1 hour. Primary antibodies (TLR7, 1:100, Abcam; pS6, 1:100, Cell Signaling; p-p65, 1:100, Cell Signaling; CD4, 1:100, Invitrogen) were incubated overnight at 4 °C. After that, sections were washed with PBS and incubated with Alexa Fluor 488- or 594-conjugated secondary antibody (Thermo Fisher Scientific) at room temperature for 1 hour. Later, sections were counterstained with 4′, 6-diamidino-2-phenylindole (DAPI) after wash. All pictures were captured using an ECLIPSE Ni-U Microscope. The fluorescence intensity among six randomly selected microscopic areas (original magnification, 200×) in each sample was measured with ImageJ.

## Cell isolation and culture

Primary human keratinocytes were isolated from human foreskin (aged 2–5) and cultured in CnT-07 (CELLnTEC, Bern, Switzerland, USA). HaCaT keratinocyte cell line obtained from NTCC (Biovector Science Lab, Beijing, China) was cultured in DMEM supplemented with 10% fetal bovine serum, penicillin–streptomycin, and 2 mM glutamine (Invitrogen). According to previous study, PBMCs were isolated from the peripheral blood of human through density gradient centrifugation (GE Healthcare) and CD4$^+$ T cells were separated by positive selection using Miltenyi beads according to the manufacturer's instructions

(Miltenyi Biotec), then cultured in RPMI 1640 medium (Gibco, Billings, MT, USA) with 10% fetal bovine serum (Biological Industries) (*Wu et al., 2018*). For R848 treatment, keratinocytes were incubated with rapamycin (50 nM, 2 h), QNZ (1 nM, 12 h) or SC75741 (20 uM, 24 h), and then were treated with R848 (10 ng/ml) for the indicated time. All experiments were performed at least three times.

## RT–qPCR
Total RNA was extracted from human skin, mice skin tissues, HaCaT keratinocytes, and primary human keratinocyte cells with TRIzol Reagent (Thermo Fisher Scientific). The quality control of RNA was conducted by a NanoDrop spectrophotometer (ND-2000, Thermo Fisher Scientific). The Maxima H Minus First Strand cDNA Synthesis Kit with dsDNase (Thermo Fisher Scientific) was applied for mRNA reverse-transcription in line with the manufacturer's instructions. RT-qPCR was carried out on a LightCycler 96 (Roche) thermocycler with iTaqTM Universal SYBR Green Supermix (Bio-Rad). Each gene's relative expression was calculated by delta-CT relative to GAPDH and the fold change was standardized into the control group.

## Enzyme-linked immunosorbent assay
The conditioned media of keratinocytes was collected after 24 h of treatment. The mice skin tissues were homogenized in the complex of RIPA buffer and protease inhibitors (Thermo Fisher Scientific). Then samples were centrifuged to obtain the supernatant at 4 °C. The production of cytokines and chemokines was measured by an enzyme-linked immunosorbent assay (ELISA), using ELISA kits for human IL-1β, human CXCL10, mouse Cxcl1 (Solarbio) and mouse Cxcl2 (Minneapolis), respectively, according to the manufacturer's instructions.

## Immunoblotting
The collected cells were dissolved in the complex of RIPA buffer and protease inhibitors (Thermo Fisher Scientific) after being washed with cold PBS. Centrifugation to obtain the supernatant and then quantifying protein with bicinchoninic acid assay (Thermo Fisher Scientific). The prepared protein samples were separated on SDS–polyacrylamide gel electrophoresis and electroblotting to PVDF membranes. Membranes were blocked in 5% non-fat milk for 1 hour at room temperature, then incubated with primary antibody at 4 °C overnight. After being washed, membranes were probed with HRP-conjugated secondary antibodies (Santa Cruz). The ChemiDocTM XRS+ system (Bio-Rad) was employed to visualize the immunoreactive bands *via* the HRP substrate (Luminata, Millipore). GE-ImageQuant LAS 4000 mini (GE Healthcare) was employed for data analysis. Quantification of pS6, p-p65 and pAKT was normalized to total S6, p65 and AKT, respectively, while quantification of other proteins was normalized to GAPDH, β-Actin or α-Tublin by densitometry. The primary antibodies are listed as follows: TLR7, 1:1000, Abcam; pS6, 1:2000, Cell Signaling; S6, 1:2000, Cell Signaling; p-p65, 1:2000, Cell Signaling; p65, 1:2000, Cell Signaling; pAKT, 1:1000, Cell Signaling; AKT, 1:1000, Cell Signaling; MYD88, 1:500, Proteintech; p100/p52, 1:2000, Cell Signaling; GAPDH, 1:20000, Proteintech; β-Actin, 1:5000, Santa Cruz; α-Tublin, 1:5000, Abcam

## Plasmid and transfection

TLR7 ectopic expression vector was generated using lentivirus vector pLVX-IRES-Neo (Addgene). Primary human keratinocytes were transfected with plasmid using Amaxa® Human Keratinocyte Nucleofector® Kit and Nucleofector® Device (LONZA) following the manufacturer's instructions, then were planted into six-well plates for additional treatments. HaCaT keratinocytes in six-well plates incubated in serum-free media were transfected with plasmid using FuGENE® HD Transfection Reagent (Promega) following the manufacturer's instructions.

## Migration assay of T cells

Primary human keratinocytes were plated at $10^5$ cells per well in 24-well plates, and stimulated with R848, QNZ or RAPA according to the indicated dosage and time. $2 \times 10^6$ purified $CD4^+$ T cells were plated in the upper transwell with 8 μm pores (Falcon) and incubated 6 hours for migration.

## Statistical analysis

Statistical analysis was conducted *via* GraphPad 8.3. Data were represented as the mean ± SEM. Normal distribution and similar variance between groups were examined. Two-tailed unpaired Student's $t$-test was for 2 groups' comparisons while 1-way analysis of variance (ANOVA) with relevant *post hoc* tests was for the comparison between multiple groups. $P$ value stands for statistical significance ($^*P < 0.05$, $^{**}P < 0.01$). Data with nonnormal distribution and different variance were analyzed with two-tailed Mann–Whitney $U$-test. Pearson's $r$-test or Spearman's $r$-test (for abnormally distributed data) was for correlation analysis.

# RESULTS

## TLR7 is overexpressed in rosacea

We collected the lesional skin biopsies in the central face of 10 rosacea patients and normal skin biopsies in the similar area of 10 healthy samples (HS), and performed RNA-sequencing to explore the pathogenesis of rosacea (*Deng et al., 2021a*). After re-analysis these data of the same participants, gene set enrichment analysis (GSEA) showed Toll-like receptor signaling pathway ranked first among the differential KEGG pathways between rosacea and HS (Figs. 1A and 1B). In addition, the fold change of TLR8 and TLR7 ranked as the top two among these TLRs, suggesting that they may take part in the procession of rosacea (Fig. 1C).

In order to figure out whether TLR8 and TLR7 act in the etiology of rosacea, we detected the expression of TLR8 or TLR7 in skin samples of participants and conducted a correlation analysis between its mRNA levels and disease severity (including CEA scores and IGA scores) among 24 patients. Although the fold change of TLR8 is more obvious than TLR7 in RNA-sequencing, there was no difference in the mRNA expression of TLR8 between rosacea patients and HS (Fig. S1A). Meanwhile, TLR8 had no correlation with CEA and IGA scores (Figs. S1B and S1C). Instead, TLR7 significantly increased in rosacea patients and was positively correlated with the IGA scores (indicative of inflammation

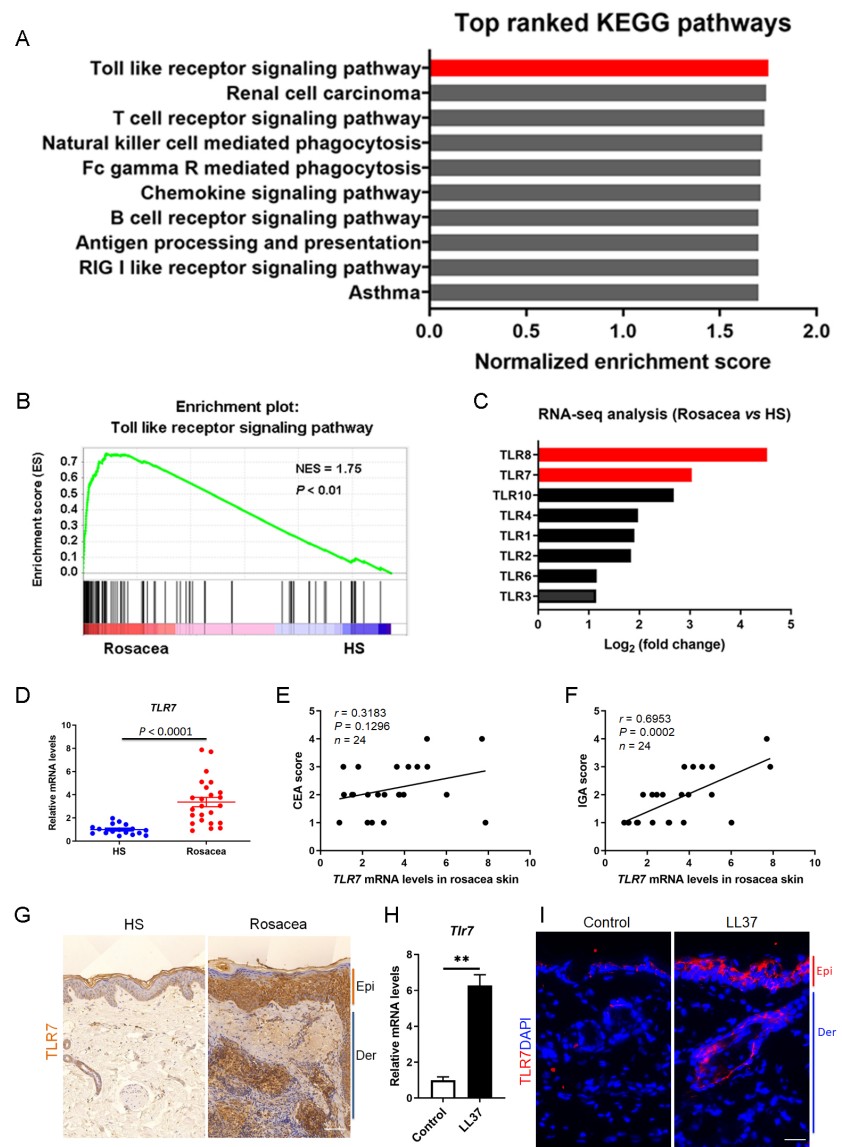

**Figure 1** **TLR7 is overexpressed in rosacea.** (A) Top-ranked upregulated KEGG terms that were differentially regulated between rosacea patients and healthy people revealed through gene set enrichment analysis (GSEA). The toll-like receptor signaling pathway was highlighted in red box. (B) GSEA on RNA-sequencing data from rosacea *versus* HS skin samples shows enrichment for the toll-like receptor signaling pathway in rosacea. HS, healthy samples; NES, normal enrichment score. Significance was calculated by permutation test. (C) The fold change of TLRs in RNA-seq analysis between rosacea and HS samples. TLR7 and TLR8 were highlighted in red box. TLRS, Toll-like receptors. (D) The mRNA expression levels of TLR7 in the epidermis of human rosacea lesions. HS, n = 17; Rosacea, n = 24. (E) Correlation of TLR7 mRNA expression in the epidermis of human rosacea lesions (n = 24) with the CEA scores. Spearman's correlation coefficient was used for the correlation analysis (two-tailed). (F) Correlation of TLR7 mRNA expression in the epidermis of human rosacea lesions (n = 24) with the IGA scores. Spearman's correlation coefficient was used for the correlation analysis (two-tailed). (G) Immunohistochemistry (IHC) of TLR7 on skin sections from HS and rosacea. Epi, epidermis; Der, dermis. Scale bar: 200 μm. (H) The mRNA expression levels of Tlr7 in skin lesions from control and LL37-induced mice (n = 6 for each group). (I) Immunostaining of TLR7 in skin lesions from control and LL37-induced mice. DAPI staining (blue) indicates nuclear localization. Scale bar: 200 μm. n = 6 for each group. Data represent the mean ± SEM. ** P < 0.01. Two-tailed unpaired Student's t-test (H) was used.

severity in rosacea) (Figs. 1D– 1F). We further detected the expression of TLR7 in the skin biopsies from rosacea patients and HS through immunohistochemistry (IHC), and found TLR7 was abundantly located in cytoplasm mainly in epidermal keratinocytes of lesional skin from rosacea patients (Fig. 1G).

Furthermore, we established a cathelicidin LL37-induced rosacea-like skin inflammation mouse model based on previous studies, which was similar to the human phenotype (*Yamasaki et al., 2007*; *Schwab et al., 2011*; *Yamasaki et al., 2011*). Mice treated with LL37 presented obvious rosacea-like clinical phenotype and pathological changes as expected (Figs. S1D and S1E). Consistent with our observation in human, the mRNA and protein expression of TLR7 in epidermis was increased remarkably in LL37-injected mice compared with control mice (Figs. 1H and 1I).

These data above suggested that TLR7, rather than TLR8, is overexpressed in the lesional skin of rosacea both in human and mice.

## TLR7 deficiency alleviates rosacea development

To determine the function of TLR7 in rosacea development, mice were intradermally administrated with scrambled (Scr) siRNA or *Tlr7* siRNA twice before injection with cathelicidin LL37 (Fig. 2A). Immunofluorescence was conducted to confirm the silencing efficacy of TLR7 in the skin in siRNA-administered mice (Fig. S2). Mice results showed that 12 hours after the last time of LL37 injection, LL37-induced mice presented typical rosacea-like skin inflammation in Scr siRNA groups while *Tlr7* siRNA groups with LL37 exhibited alleviated rosacea characteristics (Fig. 2B). Similarly, silencing TLR7 reduced the severe redness area and score caused by LL37 (Figs. 2C and 2D). Histological analysis indicated that the infiltration of inflammatory cells in the dermis induced by LL37 was improved in *Tlr7* siRNA mice compared with Scr siRNA mice (Figs. 2E and 2F). Meanwhile, RT-qPCR reflected that rosacea-related pro-inflammatory cytokines increased by LL37 were improved when epidermal TLR7 was silenced, for instance, *Tnf-α*, *Il6* and *Il-1β* (Fig. 2G). Collectively, these results proved that TLR7 deficiency could prevent the development of rosacea, which may put forward a novel treatment strategy.

## Stimulation of TLR7 activates NFκB signaling in keratinocytes

Considering that TLR7 is mainly upregulated in epidermal keratinocytes in rosacea, we transfected keratinocytes with ectopic expression vector to overexpress TLR7 to explore the underlying molecular mechanism regulating rosacea formation (Figs. S3A and S3B). The overexpression and hyperactivation of TLR7 increased the expression of its downstream adaptor protein MYD88 (Fig. S3C). We conducted RNA-sequencing from TLR7-overespressing HaCaT keratinocytes with or without R848 (a specific ligand of TLR7) treatment and identified 1,501 differentially expressed genes (DEGs) between two groups ($P < 0.05$; Fig. S3D). Gene set enrichment analysis (GSEA) exhibited that NFκB signaling pathway ranked at top two (Figs. 3A and 3B).

For the purpose of determining whether TLR7 regulates NFκB, we examined the phosphorylation level of p65/NFκB (p-p65) in TLR7-overexpressing human keratinocytes by immunoblotting, and figured out activation of TLR7 with R848 dramatically increased

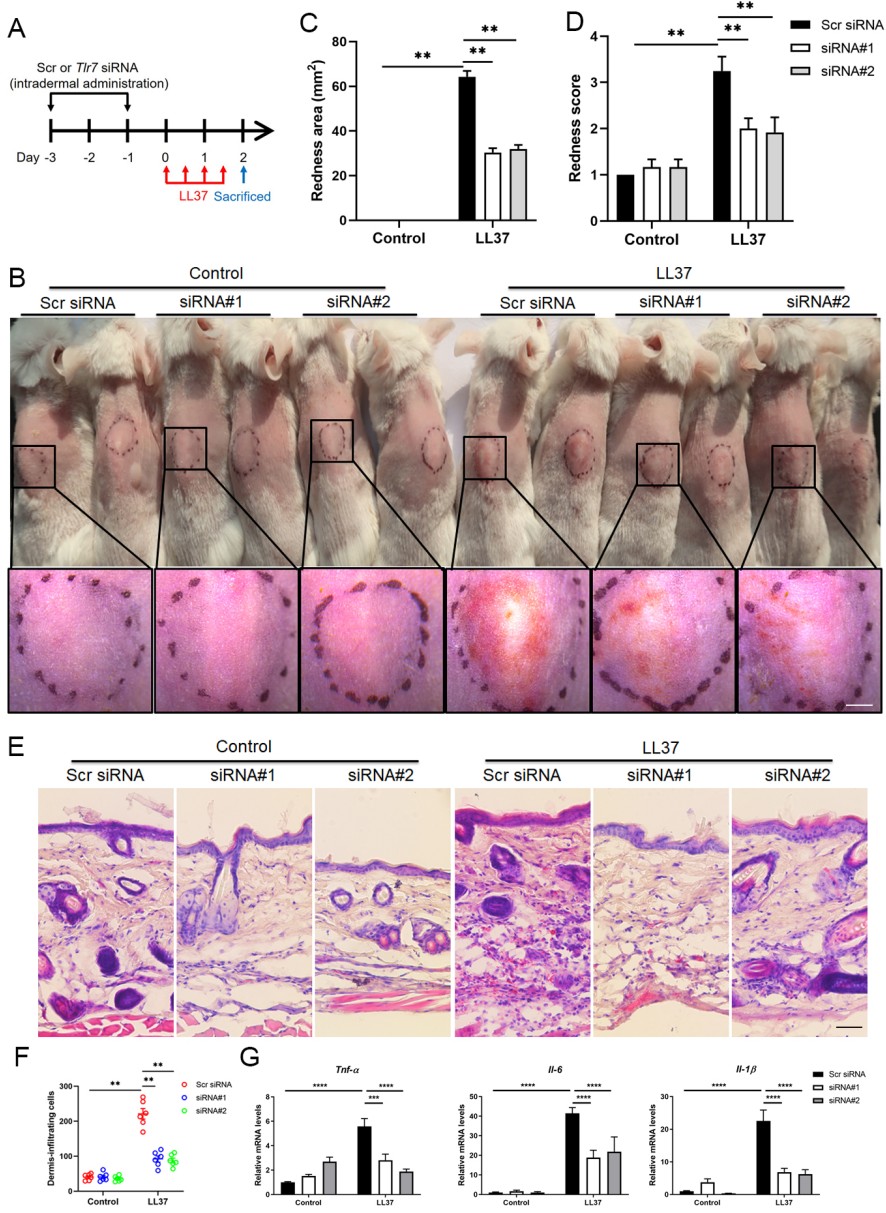

**Figure 2  TLR7 deficiency alleviates rosacea development.** (A) Schematic diagram of intraperitoneal administration of Scr siRNA or Tlr7 siRNA two times before intradermal injection of LL37 in mice. Mice were sacrificed on day 2 to conduct subsequent experiments. The mouse experiments were repeated three times, and 5–8 mice were included in each group for each time. The results of a representative mouse experiment were displayed. Scr siRNA, scrambled siRNA. (B) The back skins of Scr siRNA or Tlr7 siRNA mice were intradermally injected with LL37 or control vehicle (n = 6 for each group). Images were taken 48 h after the first LL37 administration. Below panels, magnified pictures of black boxed areas. Scale bar: 2 mm. (C) Redness area quantified to evaluate the severity of the rosacea-like phenotype (n = 6 for each group). (D) Redness score quantified to evaluate the severity of the rosacea-like phenotype (n = 6 for each group). (E) HE staining of lesional skin sections from Scr siRNA and Tlr7 siRNA mice injected with LL37 or control vehicle (n = 6 for each group). Scale bar: 200 μm. (F) Dermal infiltrating cells were quantified (n = 6 for each group) for each high power field (HPF). (G) The mRNA expression levels of Tnf-α, Il-6, Il-1β in skin lesions (n = 6 for each group). All results are representative of at least three independent experiments. Data represent the mean ± SEM. ** $P < 0.01$. One-way ANOVA with Bonferroni's *post hoc* test (C, D, F and G) was used.

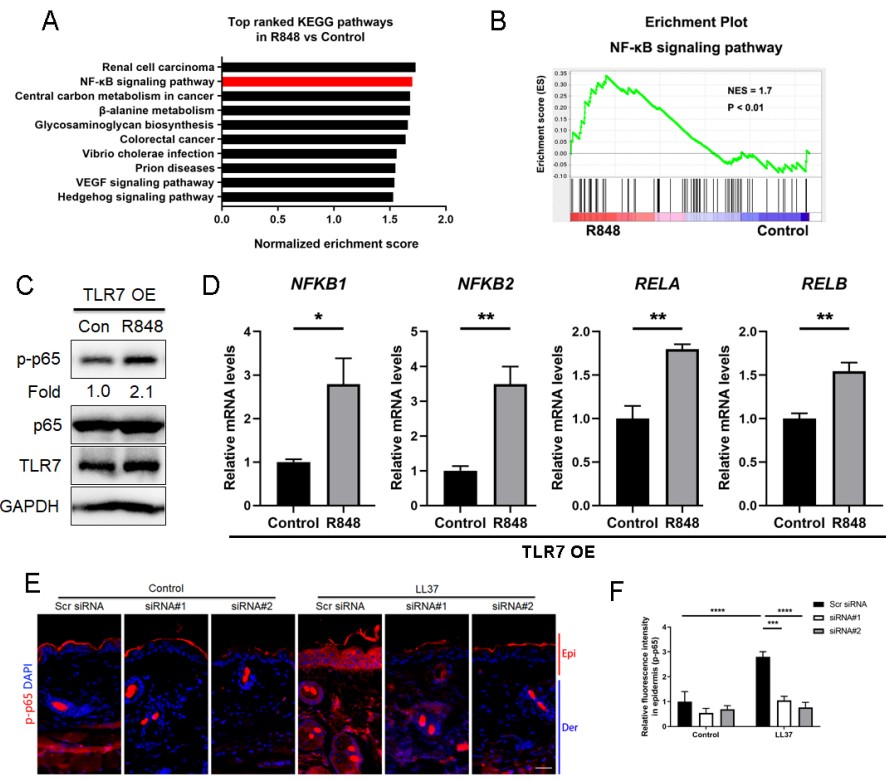

**Figure 3** **Stimulation of TLR7 activates NFκB signaling in keratinocytes.** (A) In TLR7-overexpressing HaCaT keratinocytes, top-ranked enriched KEGG terms in genes that were differentially regulated between R848 and control revealed gene set enrichment analysis (GSEA). The NFκB signaling pathway was highlighted in red box. (B) In TLR7-overexpressing HaCaT keratinocytes, GSEA on RNA-sequencing data between R848 and control shows enrichment for the NFκB signaling pathway in the R848 group. Significance was calculated by permutation test. (C) Immunoblot analysis of p-p65, total p65 and TLR7 in cell lysates from TLR7-overexpressing primary human keratinocytes treated with R848 for 15 min. p-p65 protein levels were analyzed relative to total p65. GAPDH is the loading control. Data (C) are representative of at least three independent experiments. TLR7 OE, TLR7-overexpression. (D) The mRNA expression levels of NFκB family of transcription factors (NFKB1, NFKB2, RELA, and RELB) in TLR7-overexpressing primary human keratinocytes treated with R848 for 12 h (from three replicates). (E) Immunostaining of p-p65 in skin sections from Scr siRNA and Tlr7 siRNA mice. Epi, epidermis. Der, dermis. Scale bar: 200 μm. Scr siRNA, scrambled siRNA. (F) Quantification of relative fluorescence intensity for p-p65 in epidermis (n = 6). All results are representative of at least three independent experiments. Data represent the mean ± SEM. ** $P < 0.01$. One-way ANOVA with Bonferroni's *post hoc* test (F) or two-tailed unpaired Student's t-test (D) was used.

the activation of NFκB signaling (Fig. 3C). However, the non-canonical NFκB pathway (P100/P52) can't be stimulated after hyperactivating TLR7 (Fig. S3E). Moreover, NFκB family of transcription factors (including *NFKB1*, *NFKB2, RELA*, and *RELB*) were also upregulated after R848 stimulation (Fig. 3D) (*Hiscott et al., 1993*; *Mori & Prager, 1996*). To further verify that TLR7 regulates NFκB signaling in rosacea, we performed immunofluorescence staining on mouse skin lesions and found cathelicidin LL37 increased the expression of p-p65 in epidermis, which was not apparent in *Tlr7* siRNA mice (Figs. 3E and 3F).

Taken together, this evidence explained that stimulated TLR7 in keratinocytes activates the NFκB signal pathway in rosacea.

## Hyperactivated TLR7 promotes rosacea-characteristic cytokine and chemokine production *via* NFκB signaling in keratinocytes

Since the NFκB signal pathway is closely linked to the production of inflammatory chemokines and cytokines and is reported to participate in the process of rosacea (*Deng et al., 2021a*; *Vallabhapurapu & Karin, 2009*; *Taniguchi & Karin, 2018*; *Li et al., 2022*), we detect rosacea-related cytokines and chemokines. According to the results, the expression of pro-inflammatory cytokines (*IL-1β*, *IL-6*, *IL-10* and *IL-12*) and chemokines (*CCL2*, *CCL7*, *CCL20*, *CXCL1*, *CXCL2* and *CXCL10*) raised largely in TLR7-hyperactivated keratinocytes (Fig. 4A). Meanwhile, the protein epxression of IL-1β and CXCL10 in culture supernatants was increased in TLR7-hyperactivated keratinocytes (Fig. 4B). Correspondingly, the abnormal activation of chemokines (*Ccl3*, *Ccl5*, *Ccl7*, *Cxcl1*, *Cxcl2*, *Cxcl10* and *Cxcl11* at mRNA level, as well as Cxcl1 and Cxcl2 at protein level) in mice skin resulted from LL37 was attenuated by *Tlr7* siRNA (Figs. 4C and 4D). In consideration of the chemotactic effect of chemokines on immune cells, as well as the enrichment of immune cells in rosacea lesions, we performed CD4 immunostaining and found infiltration of CD4$^+$ T cells induced by LL37 was eliminated by *Tlr7* siRNA in the dermis of rosacea-like mice (Figs. S4A and S4B). Likewise, the migration of human T cells was also promoted by TLR7-overexpressing human keratinocytes (Figs. S4C and S4E).

Next, we administrated an NFκB inhibitor QNZ (*Bottomly et al., 2022*) in TLR7-overexpressing human keratinocytes to identify the function of NFκB signaling in these processes (Fig. S4F). Not surprisingly, a number of chemokines (*IL-1β*, *CCL7*, *CCL20* and *CXCL10* at mRNA level, as well as IL-1β and CXCL10 at protein level) induced by R848 was suppressed due to attenuated NFκB signaling (Figs. 4E and 4F). Cell migration assay also demonstrated that the migrating number of human T cells induced by TLR7-overexpressing keratinocytes treated with R848 was repressed by QNZ (Figs. S4G and S4I).

To summarize, excess stimulation of TLR7 regulating rosacea-associated cytokines and chemokines to recruit T cells is dependent on NFκB signaling.

## Attenuated TLR7/NFκB signaling regulates rosacea-associated cytokine and chemokine production through inhibition of mTORC1 signaling

The mammalian target of rapamycin (mTOR), as a serine/threonine protein kinase, has been reported to promote skin inflammation in rosacea (*Deng et al., 2021a*). We detected the expression of the phosphorylated S6, the downstream molecule of mTORC1, in the epidermis of mice, and the result showed LL37 increased the expression of pS6 in Scr siRNA group but not in *Tlr7* siRNA groups (Figs. 5A and 5B). Similarly, we discovered the protein level of pS6 was also upregulated after being stimulated with R848 in TLR7-overexpressing keratinocytes (Fig. 5C), but the phosphorylation of AKT, the downstream molecule of mTORC2, remained unchanged after TLR7 activation (Fig. S5A), which confirmed the moderation effect TLR7 had on mTORC1.

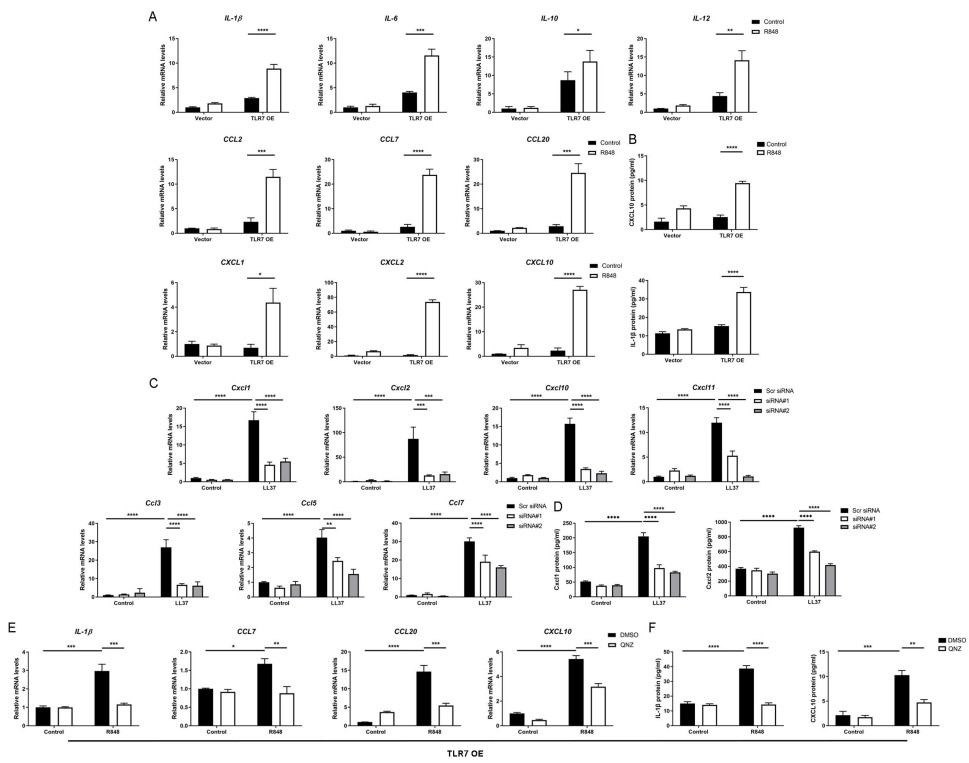

**Figure 4 Hyperactivated TLR7 promotes rosacea-characteristic cytokine and chemokine production *via* NFκB signaling in keratinocytes.** (A) The mRNA expression levels of rosacea-associated cytokines and chemokines in vector or TLR7 OE primary human keratinocytes treated with R848 (12 h) and control *in vitro*. Data represent the mean ± SEM (from three replicates). TLR7 OE, TLR7-overexpression. (B) The protein expression levels of rosacea-associated cytokine and chemokine in vector or TLR7 OE primary human keratinocytes treated with R848 (24 h) and control *in vitro* determined by ELISA. Data represent the mean ± SEM (from three replicates). (C) The mRNA expression levels of mouse chemokines in skin lesions ($n = 6$ for each group). Scr siRNA, scrambled siRNA. (D) The protein expression levels of mouse chemokines in skin lesions determined by ELISA ($n = 5$ for each group). (E) The mRNA expression levels of chemokines (*IL-1β*, *CCL7*, *CCL20* and *CXCL10*) in TLR7 OE primary human keratinocytes treated with QNZ (12 h) and then R848 (12 h) *in vitro*. Data represent the mean ± SEM (from three replicates). (F) The protein expression levels of IL-1β and CXCL10 in TLR7 OE primary human keratinocytes treated with QNZ (12 h) and then R848 (24 h) *in vitro* determined by ELISA. Data represent the mean ± SEM (from three replicates). All results are representative of at least three independent experiments. Data represent the mean ± SEM. **$P < 0.01$. One-way ANOVA with Bonferroni's *post hoc* test was used.

Numerous studies have indicated that NFκB and mTOR signals interact with each other (*Wang et al., 2017*). Consequently, we evaluated the phosphorylation of S6 after inhibiting p-p65 by QNZ in keratinocytes, and found mTORC1 was suppressed by NFκB inhibitor (Fig. 5D). This result was also confirmed by another NFκB inhibitor SC75741 (Fig. S5B) (*Reda et al., 2022*; *Zhao et al., 2022*). Curiously, when treating cells with rapamycin (RAPA), a mTORC1-specific inhibitor, the activation of NFκB cannot be repressed, which may explain the unidirectional regulatory effect of NFκB on mTORC1 (Figs. S5C and 5E).

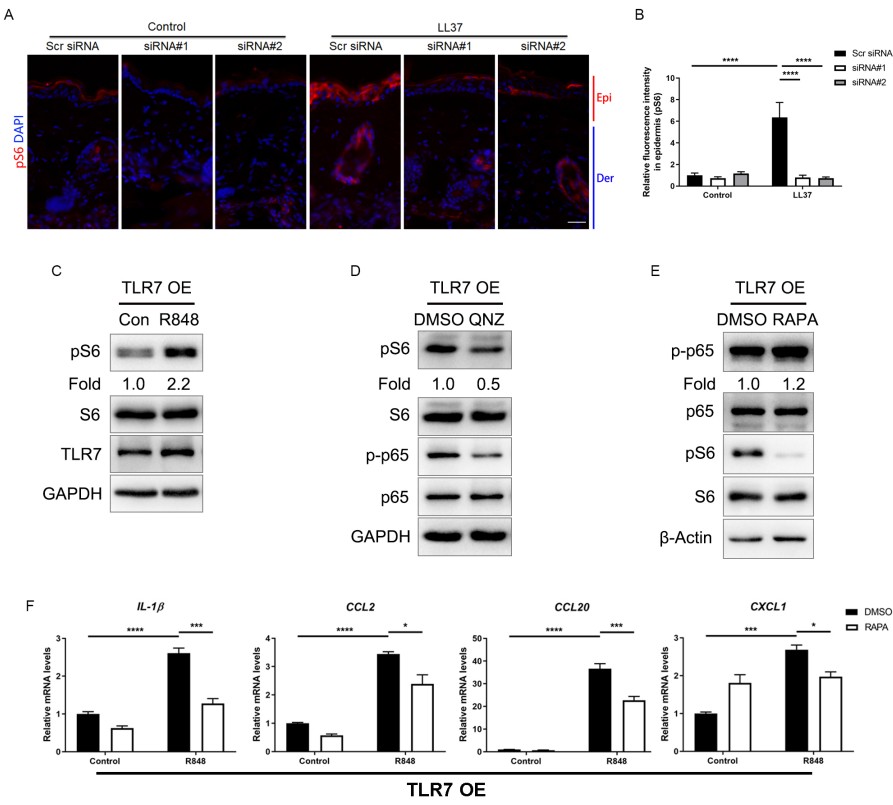

**Figure 5** **Attenuated TLR7/NFκB signaling regulates rosacea-associated cytokine and chemokine production through inhibition of mTORC1 signaling.** (A) Immunostaining of pS6 in skin sections from Scr siRNA and Tlr7 siRNA mice. Epi, epidermis. Der, dermis. Scale bar: 200 μm. Scr siRNA, scrambled siRNA. (B) Quantification of relative fluorescence intensity for pS6 in epidermis (n = 6). (C) Immunoblot analysis of pS6, total S6 and TLR7 in cell lysates from TLR7-overexpressing primary human keratinocytes treated with R848 for 15 min. pS6 protein levels were analyzed relative to total S6. GAPDH is the loading control. Data (C) are representative of at least three independent experiments. TLR7 OE, TLR7-overexpression. (D) Immunoblot analysis of pS6, total S6, p-p65 and total p65 in cell lysates from TLR7-overexpressing primary human keratinocytes treated with QNZ for 12 h. pS6, p-p65 protein levels were analyzed relative to total S6, p65, respectively. GAPDH is the loading control. Data (D) are representative of at least three independent experiments. (E) Immunoblot analysis of pS6, total S6, p-p65 and total p65 in cell lysates from TLR7-overexpressing primary human keratinocytes treated with RAPA 2 h. pS6, p-p65 protein levels were analyzed relative to total S6, p65, respectively. β-Actin is the loading control. Data (E) are representative of at least three independent experiments. (F) The mRNA expression levels of cytokines and chemokines in TLR7 OE primary human keratinocytes treated with RAPA (2 h) in vitro. Data represent the mean ± SEM (from three replicates). All results are representative of at least three independent experiments. Data represent the mean ± SEM. ** $P < 0.01$. One-way ANOVA with Bonferroni's *post hoc* test (B and F) was used.

Similar to the result of NFκB inhibition, inhibition of mTORC1 by RAPA downregulated the production of cytokines and chemokines, as well as the migration of human T cells raised by R848 (Fig. 5F, Figs. S5D–S5F).

In summary, hyperactivated TLR7/NFκB signaling may stimulate mTORC1 signaling, and encourage the production of cytokines and chemokines to recruit T cells afterward.

## DISCUSSION

In our study, we revealed TLR7 was overexpressed in rosacea, and specific knockdown of TLR7 in skin will prevent the development of rosacea in a mouse model. In human keratinocytes, hyperactivated TLR7 stimulates NFκB/mTORC1 and rosacea-characteristic cytokines and chemokines, eventually leading to skin inflammation of rosacea. These findings point out the crucial role of TLR7-NFκB-mTORC1 axis in rosacea pathogenesis.

TLR7, as a member of pattern recognition receptors, takes part in many immune-related skin diseases, including psoriasis, atopic dermatitis and so on (*Lai & Gallo, 2008*). It was commonly considered that TLR7 in the skin is expressed predominantly on immune cells such as dendritic cells (DCs) rather than keratinocytes (*Krieg, 2007*; *Bao & Liu, 2013*). Keratinocytes exert biological effects by secreting mediators to interact with TLR7 on DCs (*Tortola et al., 2012*). For example, psoriatic keratinocytes generate self-RNA, binding to TLR7 in myeloid dendritic cells to amplify the inflammatory circuits in psoriasis (*Lou et al., 2020*). However, our research provided evidence that TLR7 is dramatically overexpressed in keratinocytes in rosacea patients, which could also be induced by LL37 in mice. Besides, silencing TLR7 in keratinocytes prevented rosacea development in mice. This phenomenon illustrated that keratinocytes can sense pathogenic factors through TLR7 independently of other immune cells in the pathogenesis of rosacea, and act as immune targets and initiators to start inflammatory responses.

As "signal transducers", keratinocytes convert exogenous stimulations into cytokine and chemokine production, beginning "antigen-independent" cutaneous inflammation through a paracrine or autocrine effect (*Barker et al., 1991*). In rosacea, cytokine induction in human keratinocytes was enhanced by cathelicidin peptides, leading to augment cutaneous inflammation (*Deng et al., 2021a*; *Yamasaki et al., 2007*). Activated TLR7 produces pro-inflammatory cytokines and chemokines to regulate T cell-mediated adaptive immunity (*Bao & Liu, 2013*). In our study, TLR7 deficiency lessened inflammatory molecules induced by LL37, which inhibited the recruitment of immune cells and finally alleviated disease severity. Correspondingly, we also uncovered the expression of TLR7 in epidermis was positively associated with disease severity. Altogether, these findings illustrated the indispensable role of epidermal TLR7 in rosacea inflammatory loops.

NFκB signaling produces a marked effect on multiple inflammatory skin diseases through regulating innate and adaptive immunity (*Kang et al., 2016*; *Hara-Chikuma et al., 2015*; *Tak & Firestein, 2001*). In rosacea, the NFκB pathway promotes skin inflammation by producing excessive cytokines and chemokines (*Deng et al., 2021a*; *Chen et al., 2019*; *Li et al., 2022*; *Deng et al., 2021b*). NFκB signaling is also regulated by TLR7. The activation of the TLR7 expressed in distinct cells induces NFκB pathway, bringing about varied inflammatory or immune disorders such as lupus, atherosclerotic plaques and autoimmunity (*Guiducci et al., 2010*; *De Meyer et al., 2012*; *Demaria et al., 2010*). Mechanistically, TLR7 recruits MYD88 to associate with IL-1 receptor-associated kinase (IRAK) family members and therefore recruits TNF receptor-associated factor 6 (TRAF6) which activates NFκB (*Uematsu et al., 2005*). NFκB signaling contains two different pathways, the canonical and noncanonical (or alternative) pathways. The activation of the canonical pathway, represented by the

increased level of p-p65, regulates CD4[+] T cells through the production of cytokine production (*Liu et al., 2017*). In our study, the hyperactivation of TLR7 induced MYD88 and the canonical pathway of NFκB, resulting in the upregulation of cytokines and chemokines and enrichment of T cells. However, the underlying mechanism of how TLR7 regulates NF κB in rosacea calls for intensive research.

Previous studies demonstrated that mTOR signaling takes part in numerous cutaneous diseases, especially rosacea (*Deng et al., 2021a*; *Buerger et al., 2013*; *Naeem et al., 2017*; *Varshney & Saini, 2018*). Our experiment uncovered mTORC1 signaling stimulated by TLR7 results in cytokines/chemokines' production and T cells' enrichment in rosacea. The mTORC1/pS6 pathway plays a pivotal role in TLR signaling. Specifically, on the one hand, TLR-MYD88 complex activates PI3K-AK to phosphorylate mTOR; on the other hand, MYD88 could directly recruit phosphorylated mTOR. For this reason, IRF-5 and IRF-7 are activated and inflammatory pathway is induced (*Panda et al., 2018*). Furthermore, we found inhabitation of NFκB suppresses mTORC1 in TLR7-overexpressing keratinocytes. The interaction between NFκB and mTOR has been reported in several studies, among which, some studies claimed that NFκB transcriptionally upregulates miR-130a expression in response to inflammatory factors and finally activates mTOR signaling pathway (*Wang et al., 2017*). However, whether this hypothesis can explain our results still needs more supportive evidence.

In our previous study, hyperactivated mTOR signaling interacted with cathelicidin LL37 by binding TLR2 to induce NFκB activation and chemokines and cytokines production (*Deng et al., 2021a*). The effect of TLR2 in rosacea was clearly stated. However, the knowledge about other TLRs in rosacea is still limited. Our result showed that apart from binding TLR2, mTORC1 could also be regulated by TLR7-NFκB axis, thus produce chemokines and cytokines. This process may be independent of the positive feedback loop between mTORC1 and cathelicidin (*Deng et al., 2021a*).

Taken together, our results emphasize the key function of the TLR7 pathway in the pathogenesis of rosacea and open up new possibilities for rosacea treatment.

## ACKNOWLEDGEMENTS

We thank Prof. Lunquan Sun (Center for Molecular Medicine, Xiangya Hospital, Central South University, China) and colleagues for their generous support throughout this work.

### Funding

This work was supported by the National Key Research and Development Program of China (No. 2021YFF1201200), the National Natural Science Funds for Distinguished Young Scholars (No. 82225039), the National Natural Science Foundation of China (No. 81874251, No. 82173448, No. 82073457), Natural Science Foundation of Hunan province, China (No. 2022JJ40201). The funders had no role in study design, data collection and analysis, decision to publish, or preparation of the manuscript.

## Grant Disclosures

The following grant information was disclosed by the authors:

The National Key Research and Development Program of China: 2021YFF1201200.

The National Natural Science Funds for Distinguished Young Scholars: No. 82225039.

The National Natural Science Foundation of China: No. 81874251, No. 82173448, No. 82073457.

Natural Science Foundation of Hunan province, China: 2022JJ40201.

## Competing Interests

The authors declare there are no competing interests.

## Author Contributions

- Yaqun Huang performed the experiments, prepared figures and/or tables, and approved the final draft.
- Da Liu performed the experiments, prepared figures and/or tables, and approved the final draft.
- Mengting Chen analyzed the data, prepared figures and/or tables, and approved the final draft.
- San Xu analyzed the data, prepared figures and/or tables, and approved the final draft.
- Qinqin Peng analyzed the data, prepared figures and/or tables, and approved the final draft.
- Yan Zhu performed the experiments, prepared figures and/or tables, and approved the final draft.
- Juan Long performed the experiments, prepared figures and/or tables, and approved the final draft.
- Tangxiele Liu analyzed the data, authored or reviewed drafts of the article, and approved the final draft.
- Zhili Deng analyzed the data, authored or reviewed drafts of the article, and approved the final draft.
- Hongfu Xie conceived and designed the experiments, authored or reviewed drafts of the article, and approved the final draft.
- Ji Li conceived and designed the experiments, authored or reviewed drafts of the article, and approved the final draft.
- Fangfen Liu conceived and designed the experiments, authored or reviewed drafts of the article, and approved the final draft.
- Wenqin Xiao conceived and designed the experiments, authored or reviewed drafts of the article, and approved the final draft.

## Human Ethics

The following information was supplied relating to ethical approvals (i.e., approving body and any reference numbers):

The study were approved by the ethical committee of the Xiangya Hospital, Central South University (IRB number 201404361)

## Animal Ethics

The following information was supplied relating to ethical approvals (i.e., approving body and any reference numbers):

The study were approved by the ethical committee of the Xiangya Hospital, Central South University (201611610).

## DNA Deposition

The following information was supplied regarding the deposition of DNA sequences:

The RNA sequences are available at the Genome Sequence Archive (Genomics, Proteomics & Bioinformatics 2021) in National Genomics Data Center (Nucleic Acids Res 2022), China National Center for Bioinformation/Beijing Institute of Genomics, Chinese Academy of Sciences: GSA-Human: HRA005422.

The RNA sequences are available at Figshare: Xiao, Wenqin (2023). Sequence data. figshare. Dataset. https://doi.org/10.6084/m9.figshare.22725614.v2.

## Data Availability

The raw data are available in the Supplementary Files.

## Supplemental Information

Supplemental information for this article can be found online at http://dx.doi.org/10.7717/peerj.15976#supplemental-information.

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
