# Peer review of "TLR7 promotes skin inflammation via activating NFκB-mTORC1 axis in rosacea"

_PeerJ, doi:10.7717/peerj.15976_

## Round 0.1 · original submission · Major Revisions

Dear Dr. Huang and colleagues:

Thanks for submitting your manuscript to PeerJ. I have now received two independent reviews of your work, and as you will see, the reviewers raised some concerns about the research. In particular, it does not seem straightforward (as presented what separates this work from your prior works, and there even appears to be a similar (identical?) figure in this manuscript that was previously published by your group.

As two reviewers do not seem to be fully able to identify the new data and findings in this manuscript, I would like your revision to address this concern and CLEARLY identify what is new in this study versus prior works.

I encourage you to use the additional comments and concerns of the two reviewers to revise your work and resubmit to PeerJ. Note: all data and information should be presented, even if it was generated by prior studies. Please ensure that materials/information are provided to make your analyses repeatable. Again, new datasets (and analyses) versus published ones should be clearly noted.

I look forward to seeing your revision, and thanks again for submitting your work to PeerJ.

Good luck with your revision,

-joe

·

Basic reporting

1.Figure 1A in the present study were the same as Figure 1C in reference 13 (Deng Z, Chen M, Liu Y, Xu S, Ouyang Y, Shi W, et al. A positive feedback loop between mTORC1 and cathelicidin promotes skin inflammation in rosacea. EMBO molecular medicine. 2021;13(5):e13560), although both articles were contributed by your group.
2. What do scr siRNA, siRNA#1, siRNA#2 represent in figures?

Experimental design

3.4. hyperactivated TLR7 promotes rosacea-characteristic cytokine and chemokine porduction via NFKB signaling in keratinocytes.
All results were oringinated at RNA level, how about protein level? Are the tendencies of the protein expressions the same as those of RNA expressions?

Validity of the findings

no comment

Additional comments

no comment

Reviewer 2 ·

Basic reporting

TLR7 promotes skin inflammation via activating NFκB-mTORC1 axis in rosacea (#83734)
In this study authors have investigated the involvement of TLR7/NFkB/mTORC1 axis in promoting the production of cytokines and chemokines and observed that TLR7 is positively correlated with the inflammation and severity of rosacea. They have utilized various techniques to study human rosacea patient and a disease specific mouse model. The study is well designed and significant work is done to prove the hypothesis. Below are the specific comments which needs to be addressed before publication of the manuscript:
1. Ref. 13 “A positive feedback loop between mTORC1 and cathelicidin promotes skin inflammation in rosacea” seems to be published by the same group and has several overlapping things with this manuscript. Does the participants included in this study are same as in the previously published article? This manuscript doesn’t have patient details in a table and no mention in material and method section? Please explain?
2. RNA-Seq experiments done in this study is the same as done in the previous study? If these both analyses are different, please provide the accession number for both and if not please highlight the differences being discussed in this report and there is no data included from the previous report.
3. Authors have not discussed their above-mentioned report in this discussion properly, they should highlight that report significantly as this work seems to be the extension of the previous paper. It will lead the authors to previously published material.
4. There are several grammatical mistakes, particularly line in introduction “antimicrobial peptide LL37, and LL37 interacts with LL37 to activate mTOR signaling,” please check the whole manuscript thoroughly.

Experimental design

good

Validity of the findings

no comments

Additional comments

none

---

## Round 0.2 · Minor Revisions

Dear Dr. Huang and colleagues:

Thanks for revising your manuscript. The reviewers are very satisfied with your revision (as am I). Great! However, there are a few issues to consider. Please address these ASAP so we may move towards acceptance of your work.

Best,

-joe

·

Basic reporting

well done

Experimental design

no further comments

Validity of the findings

excellent

Additional comments

none

Reviewer 2 ·

Basic reporting

Basic reporting is ok.

Experimental design

Experimental design is explained properly but there are mistakes in writing, for example this line below
"Our previous data collected the lesional skin biopsies in"

Validity of the findings

Authors need to work on the discussion part and write at least one more paragraph explaining the new findings and how they are different from the previous study as they have re-analyzed the published data. Authors need to be very thorough with their analysis.

Additional comments

Manuscript can be accepted for publication if authors include these changes.

---

## Round 0.3 · accepted · Accept

Dear Dr. Huang and colleagues:

Thanks for revising your manuscript based on the concerns raised by the reviewer. I now believe that your manuscript is suitable for publication. Congratulations! I look forward to seeing this work in print, and I anticipate it being an important resource for groups studying immune signaling in roses. Thanks again for choosing PeerJ to publish such important work.

Best,

-joe